# Gender differences in earnings among people with multiple sclerosis and associations with type of occupation and family composition: A population-based study with matched references

**Alejandra Machado**[1]*, **Azadé Azad**[2], **Emma Pettersson**[1], **Jan Hillert**[3],
**Kristina Alexanderson**[1], **Emilie Friberg**[1]

1 Division of Insurance Medicine, Department of Clinical Neuroscience, Karolinska Institutet, Stockholm, Sweden, 2 Department of Psychology, Stockholm University, Stockholm, Sweden, 3 Division of Neurology, Department of Clinical Neuroscience, Karolinska Institutet, Stockholm, Sweden

* alejandra.machado@ki.se

**Data Availability Statement:** The highly sensitive microdata used in this study cannot be made

## Abstract

Gender differences in earnings exist worldwide. Gender segregation or familial status have been previously stated as possible explanations for these differences as well as health differences between women and men. Women are diagnosed with multiple sclerosis (MS) as twice much as men. Moreover, MS limitations may affect the work capacity of people with MS (PwMS) implying a reduction in their earnings. We aimed to explore gender differences in earnings among people with MS and without MS and between groups of those diagnosed while also considering types of occupation and family composition, and how these possible differences relate to sickness absence (SA) and disability pension (DP). We conducted a population-based cohort study in Sweden with microdata from several nationwide registers. PwMS aged 19–57 years (n = 5128) living in Sweden and 31,767 matched references from the population without MS. Outcome measures included earnings, number of SA and DP days combined (SA/DP). A four-way weighted least-squares analysis of covariance was performed to explore the associations of gender, MS, type of occupation, and family composition with earnings. Risk of SA and DP days was assessed with logistic regression. Overall, and across all occupations, women earned less than men, although less so among managers with MS. Annual gender differences in earnings were larger if living with children at home compared to not living with children. Nevertheless, these gender differences decreased after adjusting for SA/DP, both among PwMS and references. PwMS had considerably more SA/DP days than references. Women also had more SA/DP days than men. We observed that working women earned less than working men, and that gender differences in earnings were present in all occupations, although less evident among PwMS in managerial positions. The combination of gender, occupation, family composition, and MS, was associated with earnings, even when adjusting for the number of SA and DP days.

publicly available, according to several Swedish laws, such as the General Data Protection Regulation, the Swedish law SFS 2018:218, the Swedish Data Protection Act, the Swedish Ethical Review Act, and the Public Access to Information and Secrecy Act. For information about this and the data, please contact Karolinska Institutet, the Division of Insurance Medicine through professor Ellenor Mittendorfer-Rutz, co-PI of the database (contact via ellenor.mittendorfer-rutz@ki.se) and Head of the Division of Insurance Medicine.

**Funding:** This work was supported by unrestricted grants from the Swedish Social Insurance Agency (grant number: 008430-2019; awarded to EF). We used some data from the REWHARD consortium (supported by the Swedish Research Council, grant number 2017-00624). The funders had no role in study design, data collection and analysis, decision to publish, or preparation of the manuscript.

**Competing interests:** I have read the journal's policy and the authors of this manuscript have the following competing interests: AM and EF were partly funded by research grants from Biogen. EF has received unrestricted researcher-initiated grants from Bristol-Myers Squibb/Celgene. KA has received unrestricted researcher-initiated grants from Biogen. JH received honoraria for serving on advisory boards for Biogen and Novartis and speaker's fees from Biogen, Merck-Serono, Bayer-Schering, Teva, and Sanofi-Aventis. He has served as P.I. for projects sponsored by or received unrestricted research support from, Biogen, Merck-Serono, TEVA, Novartis, and Bayer-Schering. JH's MS research is also funded by the Swedish Research Council. Authors AA and EP have non-financial interests to disclose. The authors do have competing interests that alter their adherence to PLOS ONE policies on sharing data and materials.

## Introduction

The existence of earning gaps between women and men has been generally described, although these gaps are slowly decreasing [1–3]. Gender differences in earnings have been explained as the result of multiple economic and social factors that impact women's earnings, such as gender segregation of occupations [4, 5], higher levels of unpaid care work among women [6], higher childcare responsibilities [7–9], or gender-differences in morbidity [8, 10], among many other factors.

Multiple sclerosis (MS) is a chronic neurodegenerative disease, commonly diagnosed when aged 20 to 40, that is about twice as prevalent in women than in men [11, 12]. Consequently, people with MS (PwMS) are typically in the midst of other major life events when diagnosed, with significant impact on different life domains, e.g., work situation or family planning [13, 14]. Due to the progressive and unpredictable nature of MS symptoms, studies have shown a lower labour market attachment in PwMS than in the general population over time [15, 16]. In Sweden, the reduction in work capacity and thus, reduction in earnings, can be compensated with sickness absence (SA) or disability pension (DP) benefits [17–20], as well as other social protections.

In 2010, gender differences in earnings among Swedish workers (9.4%) were lower if compared to the European Union's average (12.1%). While this gender gap decreased to 7.4% in Sweden as of 2020 [3], gender differences in earnings and SA or DP benefits still exist [21], also among PwMS [16, 17, 22, 23]. Indeed a gender gap in earnings could be related to the gender segregation that still exists in many occupations in Sweden [21], affecting predominantly women in the extremely male-dominated occupations, women who also have a higher risk for SA and DP over time [4, 21]. Furthermore, part-time work is associated with these earning differences, as it is known that women work more part-time than men [21]. A common reason for working part-time is related to elderly care, childcare or being on parental leave [6, 24, 25]. Familial factors such as having underaged children or being married have also been associated with women´s earnings in general [7, 9], even in the Nordic countries [8, 25, 26]. Although these factors, such as type of occupation and family composition, have been shown to be related to the earnings of women in general as well as of women with MS, no study has investigated the interaction of these factors among PwMS and their possible associations with earnings, and if this differs between gender. Moreover, it has not been studied whether such associations differ in PwMS and others. Therefore, the aim was to explore gender differences in earnings and in number of SA/DP days among women and men diagnosed with MS and without MS. Further, to assess whether these differences are related to type of occupation and family composition among PwMS, also when considering SA/DP.

## Materials and methods

A cross-sectional nationwide cohort study was conducted, including all individuals diagnosed with MS, aged 19–57 years, and living in Sweden in 2010 and matched references from the general population without MS.

### Data

Individual-level data were linked, using the unique personal identification numbers, assigned to all residents in Sweden, and anonymized before being delivered to the researchers. Data used in the analyses were obtained from Swedish nationwide registers from the following authorities:

1. **Statistics Sweden**: *Longitudinal integration database for health insurance and labour market studies (LISA)* [27] for 2009 was used for the following variables: gender, age, educational level, country of birth (in Sweden or not), type of living area (based on population density), occupation, family composition, number of children living at home, and gross earnings. Occupations were classified based on the job title into six levels: military, managers, office work, manual labour, work not identified, and unemployed. Family factors included marital status or cohabitant and living with children under 18 years of age or not.

2. **Swedish Social Insurance Agency**: *Microdata for the analysis of the social insurance register (MiDAS)* [28] was used to obtain the sum of annual net days of sickness absence (SA) and disability pension (DP).

3. **The Swedish MS register (SMSreg)**: [29] was used to identify the PwMS diagnosed with MS in 2010 or before (8197 PwMS aged 19–57). MS duration was calculated as the time from the year of diagnosis to 2010. If information on the diagnosis year was missing (n = 786), the year of MS onset was used.

A reference group with no MS (n = 40,985) was sampled from the general population and matched on age, gender, type of living area, and county as of 31st December 2009. Five references were obtained for each MS patient (S1 Table).

Inclusion criteria were to be living in Sweden in both 2009 and 2010. Exclusion criteria were applied if not in paid work (n = 10,895; PwMS = 2758; references = 8137), if occupational classification was stated as military (<0.1% of both groups), or type of work not identified (n = 1128; PwMS = 133; references = 995). From the remaining 37,159 individuals (PwMS = 5306; references = 31,853), exclusion was also considered if already on DP for more than half-time (>183 net days) in 2009 (n = 264; PwMS = 178; references = 86). Accordingly, the final cohort included 36,895 individuals (PwMS = 5128; references = 31,767) (Fig 1).

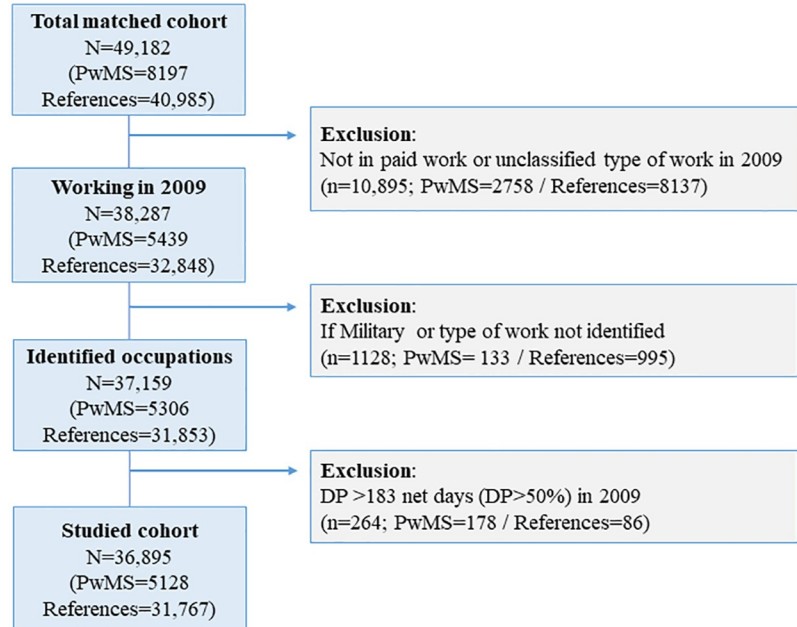

**Fig 1. Flow chart for the study sample selection.** Abbreviations: N/n, sample size; PwMS, people with multiple sclerosis; DP, disability pension.

## Sickness absence insurance in Sweden

In Sweden, all residents aged 16 years or above, with earnings or unemployment benefits are covered by the public SA insurance in case of reduced work capacity due to morbidity. For employees, the first 14 days of a SA spell are reimbursed by the employer, thereafter from the Social Insurance Agency. SA spells <15 days were not included. For individuals aged 19–64 years, DP can be granted if their work capacity is long-term or permanently reduced due to morbidity. SA covers 80% of lost income, and DP 64%, both up to a certain level. Both SA and DP can be granted for 100%, 75%, 50%, or 25% of ordinary working hours [28]. Thus, people can be on both part-time SA and DP at the same time. Therefore, net days of SA and DP were calculated (i.e., 2 days of 50% SA or DP equals 1 net day of SA/DP).

## Outcome measures

The main outcome was earnings from paid work, in the form of gross salary (before tax deductions) including possible sick pay for shorter SA spells. Another outcome was the number of SA and DP net days in 2010, combined (hereafter: SA/DP days). That is, with a maximum of 365 days, including the first 14 days of SA spells (>14 days).

## Statistical analyses

Descriptive statistics, including frequencies and percentages or mean and standard deviations for socio-demographic variables, were calculated for all PwMS and references and by gender. Distributions between both groups were tested using Chi-square (with Benjamini-Hochberg correction for multiple comparisons) and Mann-Whitney U test for continuous or skewed variables. A four-way weighted least-squares analysis of covariance (WLS ANCOVA) was conducted to compare the mean earnings in 2010 of gender (2 levels: women/men), and MS (2 levels: presence of MS = PwMS/no presence of MS = references), occupations (3 levels: manager/office/manual), and family compositions (4 categories: married or cohabitant without children/married or cohabitant with children at home/single without children/single with children at home). The model was mutually adjusted, controlling for age, type of living area (large cities/medium towns/small towns), if children <18 years living at home, and total SA/DP days for previous (2009) and studied year (2010). To adjust for the influence of outliers, the model was estimated using iteratively reweighted least squares, using Huber's M-estimator. Additionally, heteroscedasticity consistent (HC3) covariance/variance estimators were used when conducting tests and constructing confidence intervals (CI). The estimated marginal means and 95% CIs were computed for the different combinations of gender, groups, occupations, and family compositions (averaged over the control variables). Additional sensitivity analyses were performed excluding SA/DP days in 2009 and/or SA/DP days in 2010.

Multinomial logistic regression was conducted to estimate associations of gender, MS, occupation, and family composition with having 0, 1–90, and >90 SA/DP days in 2010, represented as odds ratios (OR) and 95% CIs. Statistical analyses were performed using packages from R version 4.0.4.

## Ethics approval

The project was approved by the Regional Ethical Review Board in Stockholm, Sweden (Dnr. 2007/762-31; 2011/806-32; 2014/236-32) who waived need for informed consent [30]. The study was performed in accordance with the Declaration of Helsinki and later amendments. The data were obtained from total population registers kept by Swedish administrative authorities and all data were delivered anonymized to us as researchers. Individuals with MS included

in the Swedish MS Registry (SMSreg) were given written and verbal information prior to the inclusion to the register. MS patients also provided verbal consent to their neurologist to enter their information into the register for both clinical and research purposes. This is documented in the register together with the date of consent and the person who introduced this information as a prove of witness. Informed consent from the research participants was not applicable due to the use of pseudonymized data from total population administrative registers and that we do not hold the details revealing the identity of the participants.

## Results

The final cohort consisted of 5128 PwMS (71.4% women) and 31,767 matched references (71.7% women). After exclusion criteria were applied, the original balance produced by exact matching was no longer preserved for age and type of living area. The distribution of county was not affected ($\chi2 = 0.200$; p = 1.000) and gender proportions were also similar between groups: PwMS and references ($\chi2 = 0.122$; p = 0.726), with ~71% of women in both groups. Furthermore, there was no gender difference regarding MS disease duration. Differences between groups' proportions were present for the other sociodemographic variables (p≤0.001) (Table 1).

As a result of unbalanced cohort matching and the presence of moderate outliers in earnings, a WLS ANCOVA was conducted to estimate differences in mean earnings in 2010 (in Swedish Krona, SEK) across the four independent variables (gender, MS, occupations, and family composition), and adjusting for age, type of living area, number of children living at home, and mean annual SA/DP days in 2009 and 2010. Results from the four-way and three-way interaction between gender, MS, occupation, family composition were significant (Table 2). Regarding the two-way interactions, all combinations except those including the MS variable were significant. Similarly, all main effects but MS, as well as all control variables (except for number of children at home), had a significant association with earnings (Table 2). Due to the wide-ranging results from the factorial ANCOVA model, apart from the four-way interaction, we focused mainly on the results related to the interactions including gender and MS, the combined interaction of gender, MS, and occupation, as well as gender, MS, and family composition (see S2 Table for further details).

### Gender differences in earnings among people with or without MS

Gender differences in earnings were present. Averaged over all model covariates, the estimated difference between women's and men's earnings in 2010 was: 63,884.93 SEK (95% CI: 56,561.95–71,207.91) or 7,091.2 Euros. Importantly, this difference has significantly different values at the varying levels of MS, family composition, and occupation. The interaction of gender and MS has a significantly higher-order effect on occupation or family composition (Table 2). However, the pattern of gender differences in earnings among PwMS and references and across family composition and occupations was somewhat heterogeneous (Fig 2). Gender differences in earnings were predominant among office and manual workers in all family compositions, where women on average earned significantly less than men in similar occupations or family composition, both among PwMS and references. The only exceptions were single office and manual workers with MS living with children; these women and men had, to some extent, similar mean earnings in 2010 (Fig 2- centre & left). There were distinct gender differences in earnings among managers. Among PwMS, differences in earnings were observed only among the managerial women and men who were single without children at home; these women earned significantly less than these men (Fig 2- left). In contrast, among references,

**Table 1. Descriptive characteristics for sociodemographic and clinical variables in 2009 for people with MS (PwMS) and matched references, by gender and among all.**

| | People with MS | | | Reference group (No MS) | | |
|---|---|---|---|---|---|---|
| Gender | Women | Men | All with MS | Women | Men | All references |
| | 3663 (71.4) | 1465 (28.6) | 5128 (100) | 22767 (71.7) | 9000 (28.3) | 31767 (100) |
| **Age [a,c]** | | | | | | |
| 19–24 | 95 (2.6) | 30 (2) | 125 (2.4) | 467 (2.1) | 155 (1.7) | 622 (2) |
| 25–34 | 742 (20.3) | 292 (19.9) | 1034 (20.2) | 3783 (16.6) | 1466 (16.3) | 5249 (16.5) |
| 35–44 | 1318 (36) | 586 (40) | 1904 (37.1) | 7717 (33.9) | 3303 (36.7) | 11020 (34.7) |
| 45–54 | 1232 (33.6) | 446 (30.4) | 1678 (32.7) | 8564 (37.6) | 3198 (35.5) | 11762 (37) |
| 55–57 | 276 (7.5) | 111 (7.6) | 387 (7.5) | 2236 (9.8) | 878 (9.8) | 3114 (9.8) |
| Mean age (Sd) [a] | 41.82 (8.9) | 41.62 (8.7) | 41.76 (8.8) | 43.15 (8.8) | 42.95 (8.7) | 43.1 (8.8) |
| **Educational level [a,b,c]** | | | | | | |
| Compulsory school (<10 years) | 190 (5.2) | 151 (10.3) | 341 (6.6) | 1663 (7.3) | 1160 (12.9) | 2823 (8.9) |
| Upper secondary school (10–12 years) | 1598 (43.6) | 742 (50.6) | 2340 (45.6) | 10530 (46.3) | 4632 (51.5) | 15162 (47.7) |
| University/college (>12 years) | 1875 (51.2) | 572 (39) | 2447 (47.7) | 10574 (46.4) | 3208 (35.6) | 13782 (43.4) |
| **Born in Sweden [a]** | | | | | | |
| Yes | 3413 (93.2) | 1346 (91.9) | 369 (7.2) | 19798 (87) | 7889 (87.7) | 4080 (12.8) |
| No | 250 (6.8) | 119 (8.1) | 4759 (92.8) | 2969 (13) | 1111 (12.3) | 27687 (87.2) |
| **Type of living area [a]** | | | | | | |
| Larger cities | 1511 (41.3) | 632 (43.1) | 2143 (41.8) | 8839 (38.8) | 3618 (40.2) | 12457 (39.2) |
| Medium-sized towns | 1259 (34.4) | 484 (33) | 1743 (34) | 7871 (34.6) | 3138 (34.9) | 11009 (34.7) |
| Small towns | 893 (24.4) | 349 (23.8) | 1242 (24.2) | 6057 (26.6) | 2244 (24.9) | 8301 (26.1) |
| **Family composition [a,b,c]** | | | | | | |
| Married/cohabitant, no children at home | 741 (20.2) | 232 (15.8) | 973 (19) | 4776 (21) | 1409 (15.7) | 6185 (19.5) |
| Married/cohabitant with children at home | 1518 (41.4) | 632 (43.1) | 2150 (41.9) | 9704 (42.6) | 4063 (45.1) | 13767 (43.3) |
| Single no children at home | 1139 (31.1) | 559 (38.2) | 1698 (33.1) | 6167 (27.1) | 3280 (36.4) | 9447 (29.7) |
| Single with children at home | 265 (7.2) | 42 (2.9) | 307 (6) | 2120 (9.3) | 248 (2.8) | 2368 (7.5) |
| **Occupation [a,b,c]** | | | | | | |
| Manager | 140 (3.8) | 106 (7.2) | 246 (4.8) | 1124 (4.9) | 872 (9.7) | 1996 (6.3) |
| Office | 2241 (61.2) | 699 (47.7) | 2940 (57.3) | 12022 (52.8) | 3747 (41.6) | 15769 (49.6) |
| Manual | 1282 (35) | 660 (45.1) | 1942 (37.9) | 9621 (42.3) | 4381 (48.7) | 14002 (44.1) |
| **MS Disease duration (years)**, M(sd) | 7.45 (6.3) | 7.29 (6.4) | 7.4 (6.4) | - | - | - |

Note: the variable County was included in the analysis but not displayed in this table. For further information on this variable please see S1 Table.

Proportion differences were assessed with the Chi-square test and adjusted for all pairwise comparisons using the Benjamini-Hochberg correction. Group comparisons of mean age and mean disease duration were tested with the Mann-Whitney test. Significant results are based on two-sided tests (p≤0.001):

[a] Proportional or mean difference between groups (PwMS vs References, shaded in light grey)

[b] Proportional or mean differences between women and men among PwMS

[c] Proportional or mean differences between women and men among the reference group.

the gender gap in earnings was present in all family compositions except among those single without children at home, where the mean earnings were similar.

## Gender differences in earnings in association with MS, occupation, or family composition

Differences in earnings among PwMS and references were present although not at all levels (Table 2). However, the interaction of MS with gender, type of occupation, and/or family composition had a significant impact on earnings (Table 2, Three- and Four-way interactions). For

**Table 2. Four-way weighted least-squares analysis of covariance (WLS ANCOVA) main effects and interactions on mean earnings.**

|  | Df | F | p-value |
|---|---|---|---|
| **Main effects** |  |  |  |
| Gender | 1 | 341.6 | **<0.001** |
| MS | 1 | 0.2 | 0.651 |
| Occupation | 2 | 523.2 | **<0.001** |
| Family composition | 3 | 31.2 | **<0.001** |
| **Two-way interactions** |  |  |  |
| Gender x MS | 1 | 2.8 | 0.095 |
| Gender x Occupation | 2 | 10.4 | **<0.001** |
| Gender x Family composition | 3 | 65.4 | **<0.001** |
| MS x Occupation | 2 | 1.2 | 0.298 |
| MS x Family composition | 3 | 0.9 | 0.413 |
| Occupation x Family composition | 6 | 4.8 | **<0.001** |
| **Three-way interactions** |  |  |  |
| Gender x MS x Occupation | 2 | 3.9 | **0.020** |
| Gender x MS x Family composition | 3 | 3.3 | **0.019** |
| Gender x Occupation x Family composition | 6 | 3.6 | **0.001** |
| MS x Occupation x Family composition | 6 | 3.4 | **0.002** |
| **Four-way interaction** |  |  |  |
| Gender x MS x Occupation x Family composition | 6 | 4.3 | **<0.001** |
| **Covariates** |  |  |  |
| Age | 1 | 1848.8 | **<0.001** |
| Type of living area | 2 | 240.1 | **<0.001** |
| Children under 18 living at home | 1 | 0.8 | 0.377 |
| SA/DP days in 2009 | 1 | 61.7 | **<0.001** |
| SA/DP days in 2010 | 1 | 623.2 | **<0.001** |

Abbreviations: MS, multiple sclerosis; Df, degrees of freedom; F, F statistic

Note: Analysis of variance table (Type III SS, HC3 heteroscedasticity corrected covariance matrix used). Significant results are marked in bold (p<0.05).

Factor levels: gender (2 levels: women/men), and MS (2 levels: MS = PwMS/no MS = references), different occupations (3 levels: manager/office/manual) and family compositions (4 categories: married or cohabitant without children/married or cohabitant with children at home/single without children/single with children at home).

example, PwMS and references in office and manual work had similar mean annual earnings. The only exception was for male office workers married/cohabitant with children at home, where male references earned significantly more than men with MS (Fig 2- centre). Again, a different pattern was observed among managers. Men with MS in managerial positions who were married/cohabitant (with or without children) earned significantly less than men without MS in a similar situation. Contrary, managerial women with MS who were single with no children at home had significantly lower mean annual earnings than managerial women with no MS in the same circumstances (Fig 2- left).

Furthermore, there was a significant association between the type of occupation and levels of earnings. Managers earned more than office workers who in turn earned more than manual workers, irrespective of being diagnosed with MS or not (Fig 2). Gender differences were present within each occupation, both for PwMS and references. The main and higher-order effects of family composition on earnings were also significant, however, no clear pattern could be

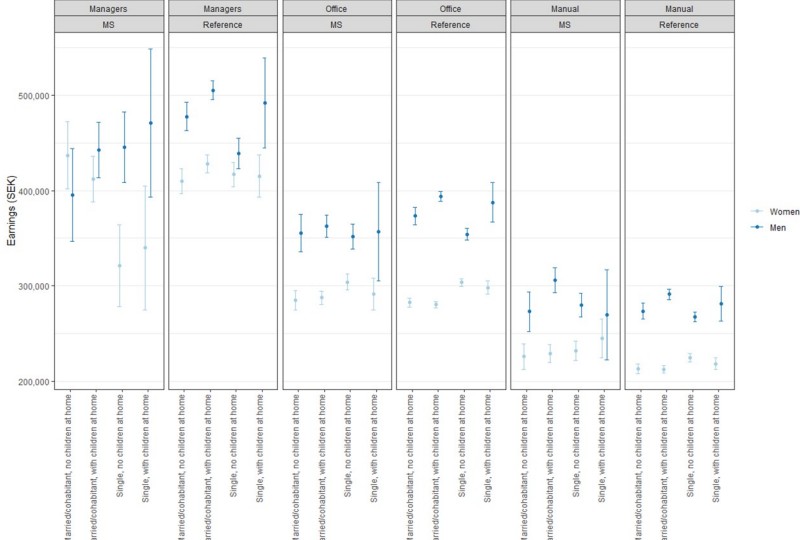

**Fig 2. Four-way weighted least-squares analysis of covariance (WLS ANCOVA) plots for mean earnings (in Swedish Krona, SEK) between gender (women, in light blue vs men, in dark blue), presence of MS or not (MS vs references), occupation (managerial, office, or manual workers), and family composition (married/cohabitant or single, and with or without children <18 years at home, respectively) when mutually adjusted and controlling for the effects of age, type of living area, if children under 18 living at home, and total net days of SA/DP for 2009 and 2010.**

depicted when family composition was combined with other factors (i.e., higher or lower mean annual earnings varied across family composition, occupations, gender, and MS). Nevertheless, a trend for higher mean earnings was observed among male workers who were married/cohabitant with children at home, irrespective of MS or not (Fig 2). No clear trend was found among women, where mean earnings were somewhat comparable across all levels of family compositions among women with and without MS when in office or manual work.

Additionally, sensitivity analyses were conducted (Table 3; all models). The factorial model was reproduced by adjusting for age, type of living area, number of children <18 living at home, and SA/DP days in 2009 (model 2) and only for age and type of living area (model 3). Overall, the interaction of all four factors (gender, MS, type of occupation, and family composition) on earnings remained significant when not adjusting for number of children living at home, SA/DP days in 2009 and 2010 (model 3), and also when excluding age and type of living area (model 4). Importantly, the confounding variables for SA/DP of 2009 and/or 2010 had a clear covariate effect over other factors (models 1 and 2), mainly on the interactions of MS with gender, occupation, or family composition (Table 3). For example, number of SA/DP days in 2009 and/or SA/DP in 2010 may explain the differences in earnings between PwMS and references. However, after adjusting, significant earning differences were still present in several interactions: "Gender x MS x Occupation", "Gender x MS x Family composition", and "Gender x Occupation x Family composition" (Table 3).

In summary, similar patterns of gender difference in earnings were observed in all 4 models —i.e., for both groups, and across all types of occupation and family compositions, women had lower mean annual earnings than men. Moreover, when not adjusting for SA/DP days in 2009 and/or SA/DP days in 2010, a broader gender earning gap among PwMS (in all levels of family composition and occupations) was observed than among references (see S1 Fig for WLS ANCOVA plots of model 3). Results showed the same decreasing pattern in levels of

**Table 3. Sensitivity analysis of a four-way weighted least-squares analysis of covariance (WLS ANCOVA) main effects and interactions on mean earnings in 2010 when mutually adjusted and controlling for distinct confounder variables.**

|  | Model 1 | Model 2 | Model 3 | Model 4 |
|---|---|---|---|---|
|  | p-value | p-value | p-value | p-value |
| **Main Effects** |  |  |  |  |
| Gender | **<0.001** | **<0.001** | **<0.001** | **<0.001** |
| MS | 0.651 | 0.273 | **<0.001** | 0.773 |
| Occupation | **<0.001** | **<0.001** | **<0.001** | **<0.001** |
| Family composition | **<0.001** | 0.188 | **<0.001** | 0.728 |
| **Two-way interactions** |  |  |  |  |
| Gender x MS | 0.095 | **0.001** | 0.290 | **<0.001** |
| Gender x Occupation | **<0.001** | **<0.001** | **<0.001** | **0.002** |
| Gender x Family composition | **<0.001** | **<0.001** | **<0.001** | **<0.001** |
| MS x Occupation | 0.298 | 0.184 | **0.006** | **0.006** |
| MS x Family composition | 0.413 | **<0.001** | **<0.001** | **<0.001** |
| Occupation x Family composition | **<0.001** | **<0.001** | **<0.001** | **<0.001** |
| **Three-way interactions** |  |  |  |  |
| Gender x MS x Occupation | **0.020** | **0.032** | **0.002** | **<0.001** |
| Gender x MS x Family composition | **0.019** | **<0.001** | 0.133 | **<0.001** |
| Gender x Occupation x Family composition | **0.001** | **0.002** | **0.002** | **0.001** |
| MS x Occupation x Family composition | **0.002** | **0.001** | **<0.001** | **<0.001** |
| **Four-way interaction** |  |  |  |  |
| Gender x MS x Occupation x Family composition | **<0.001** | **<0.001** | **<0.001** | **<0.001** |
| **Covariates** |  |  |  |  |
| Age | **<0.001** | **<0.001** | **<0.001** | - |
| Type of living area | **<0.001** | **<0.001** | **<0.001** | - |
| Children under 18 living at home* | 0.377 | 0.476 | - | - |
| SA/DP days in 2009 | **<0.001** | **<0.001** | - | - |
| SA/DP days in 2010 | **<0.001** | - | - | - |

Abbreviations: MS, multiple sclerosis; Df, degrees of freedom; F, F statistic

Note: Analysis of variance table (Type III SS, HC3 heteroscedasticity corrected covariance matrix used). Significant results are marked in bold (p<0.05).

*Model 1*: Mutually adjusted factorial model controlling for age, type of living area, children under 18 living at home, SA/DP days in 2009 and SA/DP days in 2010.

*Model 2*: Mutually adjusted factorial model controlling for age, type of living area, children under 18 living at home, SA/DP days in 2009.

*Model 3*: Mutually adjusted factorial model controlling for age and type of living area.

*Model 4*: Unadjusted factorial model

\* The variable number of children living at home was excluded from models 3 and 4 as this variable was not significant in models 1 or 2.

earning related to type of occupation (i.e., higher earnings among managerial positions, followed by office and manual workers) across gender and MS. However, in comparison, to the adjusted model shown in Fig 2, these gender differences in earnings within the same type of occupation were larger (S1 Fig).

## SA/DP days

A multinomial logistic regression was performed to examine the associations between having different numbers of SA/DP days in 2010 and our studied factors: gender, MS, occupation, and family composition, whilst adjusting for age and type of living area. We calculated the ORs of having 1–90 SA/DP days and 91–365 SA/DP days, respectively, compared to having no SA/DP days, for all levels of the covariates (see Table 4). Overall, and when compared to

**Table 4. Odds ratios (OR) with 95% confidence intervals (CI) for having a different number of sickness absence and disability pension net days in 2010, compared to references, categorized as 0, 1–90, and >90 days.**

| | SA/DP net days | |
|---|---|---|
| | OR (95% CI) | OR (95% CI) |
| | *0 vs 1–90* | *0 vs 91–365* |
| **Age** | **1.05 (1.04–1.07)** | **1.09 (1.08–1.09)** |
| **Gender** | | |
| Men | ref | ref |
| Women | **1.76 (1.37–2.26)** | **2.26 (2.01–2.55)** |
| **Type of living area** | | |
| Larger cities | ref | ref |
| Medium-sized towns | 1.24 (0.98–1.57) | **1.44 (1.29–1.61)** |
| Small towns | **1.35 (1.05–1.75)** | **1.64 (1.45–1.84)** |
| **MS** | | |
| References | ref | ref |
| PwMS | **15.72 (12.76–19.37)** | **23.40 (21.17–25.86)** |
| **Family composition** | | |
| Married/cohabitant, no children at home | ref | ref |
| Married/cohabitant with children at home | 1.05 (0.78–1.42) | 1.07 (0.93–1.22) |
| Single no children at home | **1.30 (1.01–1.66)** | **1.30 (1.16–1.46)** |
| Single with children at home | **1.65 (1.15–2.39)** | 1.19 (0.98–1.44) |
| **Occupation** | | |
| Manager | ref | ref |
| Office | **1.74 (1.00–3.01)** | **2.25 (1.70–2.98)** |
| Manual | **2.03 (1.16–3.54)** | **3.33 (2.51–4.42)** |

Abbreviations: SA, sickness absence; DP, disability pension; OR, odds ratios; CI, confidence intervals; ref, reference. The model is mutually adjusted for age, gender, and type of living area. Significant results are marked in bold (p<0.05).

references, PwMS were more likely to have 1–90 SA/DP days (OR = 15.82; 95% CI:12.76–19.37) and >90 SA/DP days (OR = 23.40; 95% CI:21.17–25.86). Moreover, higher ORs of having 1–90 days or more (91–365 days) were associated with higher age, being women, living in small towns, being single with no children at home, and having an office or manual occupation. In addition, single parents showed a higher risk of having 1–90 SA/DP days, and those living in medium-sized towns had a higher risk for >90 SA/DP days compared to those living in big cities (Table 4).

## Discussion

The explorative results of this nationwide cohort study of people with and without MS showed that, women earned less than men, and that the gender differences in earnings were present in all types of occupations, although less evident among PwMS in managerial positions. Moreover, the combination of gender, type of occupation, family composition, and having an MS diagnosis, was significantly associated with earnings, even when adjusting for number of sickness absence and disability pension days. Furthermore, across all occupations, and predominantly among references, the gender gap in earnings was even larger when living with underaged children in comparison to not living with children. Finally, PwMS had significantly higher SA/DP days compared to the matched references without MS. Women also had

significantly higher SA/DP days compared to men. When not considering the influence of SA/DP days, gender gaps increased to some extent, mainly among PwMS.

Gender differences in earnings exist worldwide [1–3]. While these differences have decreased substantially over time, they are still present [1–3]. Several studies suggest that gender differences in earnings could be related to the existing numerical gender segregation of the labour market [4, 5], higher childcare responsibilities [7–9], or to gender differences in morbidity [8, 10]. The presence of a chronic disease like MS is known to have a direct consequences on earnings as function limitations due to the disease are known to reduce work capacity over time [11, 17, 23].

Our findings, that gender differences in earnings were present in all types of occupations, are in line with official statistical reports from the year 2011 in Sweden, reflecting that, even female-dominated occupations, women earned less than men for that same profession [31]. Our findings also showed an association between type of occupation and level of earnings, both for PwMS and references, with a gradient in earnings between managers, office workers, and manual labourers, in line with other reports [21, 31]. A previous study reported this gradient in earnings related to the type of occupation in PwMS [17], but also equivalent earnings between PwMS and their references when in more qualified occupations [17]. Moreover, the type of occupation has previously shown to play a role in the level and trajectories of SA and DP overtime, with managers showing lower levels of SA/DP days compared to other types of occupations, [32] and a higher risk of SA/DP days for office and manual occupations [17, 32]. In line with two previous studies, our findings indicate that the type of occupation or educational level are influential determinants of the large heterogeneity of PwMS' earnings and SA/DP days [17, 32]. Moreover, more SA/DP days were found among women compared to men, as well as among PwMS compared to references [28, 33]. Altogether, this could also explain why the earnings difference between women and men as well as between PwMS and references, was larger when not adjusting for SA/DP days in 2010 and 2009.

While evident differences in levels of earnings were observed among all three types of occupations as well as gender differences in earnings within the same type of occupation (in most cases), this gender gap varied when associated with different levels of family composition. For example, women earned less on average than men when living with children, irrespective of the type of occupation or presence of MS or not. This finding could reflect what other authors have found in relation to the impact of parenthood on women's earnings due to temporal absence of work participation or reduced working hours [24, 25]. For example, Kleven et al. [34] showed that gender differences in earnings in Denmark was attributed to the dynamic effects of children; mothers earned 20% less than fathers. However, these authors specified that several other reasons may affect this earning difference, such as labour force participation (full- or part-time), occupation, sector, and environmental influences (i.e., female and male gender identity formed during childhood) when choosing between family or a professional career [34]. Such results converge, to a certain extent, with our findings of married/cohabitant men with children having higher earnings across all occupations, a trend that even remained when controlling for annual SA/DP days.

Moreover, the combined association of type of occupation, family composition, and MS also showed a clear contribution to gender differences in earnings, which can vary across all factor combinations. For example, the mean earnings among managerial women with MS differed from managerial women with no MS. While managerial women without MS had similar earnings irrespective of family composition, managerial women with MS had lower mean earnings when single—irrespective of living with children or not, compared to those married or cohabitating with or without children. This finding may point towards the flexibility of the managerial occupation itself which could allow mothers in managerial and professional

occupations to have similar earnings to their counterparts without children [35]. On the other hand, lower mean earnings among managerial single women with MS could also indicate that some of them work part-time, as it is known that a higher proportion of individuals with MS are in part-time employment compared to in the general population, at least outside of Sweden [36]. To what extent this is the case in Sweden among women with MS in general and among those living with children, should be investigated in future studies. The same goes for willingness to undertake a managerial position when also handling possible MS limitations.

Another aspect, however, not studied here, is the large numerical gender segregation of occupations in Sweden and other countries [4, 5]. Most occupations are either male- or female-dominated and there are very few gender integrated occupations. Salaries tend to be higher in the male-dominated occupations; however, also among men in the female-dominated occupations. Hence, despite that the employment frequency among women in Sweden is one of the highest worldwide [21], the numerical gender segregation across occupations could probably explain some of the observed gender differences in earnings [4, 5]. We suggest that this is explored in future studies.

The strength of this study lies in the large MS cohort and matched references, based on the use of comprehensive high-quality nationwide register data [28, 29, 37], with high validity and no drop-outs. Another strength is the very high female employment frequency in Sweden [21]. Moreover, the statistical approach used in this study allowed to investigate the effect of multiple independent variables in the dependent variable at once, whilst controlling for the effect of known covariates. This factorial ANCOVA is supposed to explore the data more efficiently, mainly in the presence of several factors of interest.

The study also has some limitations that need to be addressed, limitations that means that some results must be considered with caution. First, as this is a cross-sectional study, directions of identified associations are not possible to determine. Second, we had no information about to what extent people worked full- or part-time, or worked only part of 2010, e.g., due to parental leave, studying, unemployment, etcetera. In that case their earnings would have been lower. Nevertheless, we attempt to control for part-time work for those with different levels of SA/DP days, which would mostly affect possible differences in earnings between PwMS and references, as shown from WLS ANCOVA models before and after adjusting for SA/DP days. Further, the sample sizes for managerial occupations for both women and men who were single and living with children were small compared to the other subgroups, as reflected by large CIs. Thus, results need to be interpreted with caution. Fourth, some cohabitants without children might have been registered as singles. Finally, the breadth of the occupational groups could disguise an already existent unbalance between inherent female vs male-dominated occupations in mean earnings. Further studies are suggested to unravel these possible influences.

## Conclusions

Overall, we found that working women earned less than working men in general as well as among PwMS. The individual and combined associations with type of occupation and family composition among the PwMS entail unequal gender earning profiles affecting most working women. Nevertheless, the identified gender difference in earnings decreased when adjusting for previous and current SA/DP. Although these results cannot draw causal associations, it can certainly offer several insights for future studies. Primarily, as there is a need for studies focused on family, work, and income of women and men with MS, as they have been shown to be important life domains affecting the quality of life, psychosocial adjustment, and health outcomes in PwMS.

## Supporting information

**S1 Fig. Four-way weighted least-squares analysis of covariance (WLS ANCOVA) plots for mean earnings (in Swedish Krona, SEK) between gender (women, in light blue vs men, in dark blue), presence of MS or not (MS vs references), occupation (managerial, office, or manual workers), and family composition (married/cohabitant or single, and with or without children <18 years at home, respectively) when mutually adjusted and controlling for age and type of living area only.**
(PDF)

**S1 Table. Sociodemographic, economic, and clinical characteristics in 2009 of people with multiple sclerosis (PwMS) and of matched references, by gender and among all.** Abbreviations: MS, Multiple sclerosis; Sd, Standard deviation; SEK, Swedish Krona; SA, Sickness absence; DP, disability pension. Both cohorts were matched on age, gender, type of living area, and county with a ratio of 5 reference individuals sampled from the general population to one person with multiple sclerosis (MS).
(PDF)

**S2 Table. Regression estimates (Robust (weighted) regression, with heteroscasticity corrected (HC3) standard errors/confidence intervals).** Abbreviations: CI, Confidence interval; p: p values. Significant results are marked in bold (p<0.05).
(PDF)

## Author Contributions

**Conceptualization:** Alejandra Machado, Azadé Azad, Emilie Friberg.

**Formal analysis:** Alejandra Machado, Emma Pettersson.

**Funding acquisition:** Kristina Alexanderson, Emilie Friberg.

**Methodology:** Alejandra Machado, Emma Pettersson.

**Project administration:** Alejandra Machado, Emilie Friberg.

**Software:** Emma Pettersson.

**Supervision:** Emilie Friberg.

**Writing – original draft:** Alejandra Machado, Azadé Azad.

**Writing – review & editing:** Alejandra Machado, Azadé Azad, Emma Pettersson, Jan Hillert, Kristina Alexanderson, Emilie Friberg.

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
