## [Decision Letter · Decision Letter 0]

7 Jun 2023

PONE-D-22-32848Gender differences in earnings among people with multiple sclerosis and associations with type of occupation and family composition: a population-based study with matched referencesPLOS ONE

Dear Dr. Machado,

Thank you for submitting your manuscript to PLOS ONE. After careful consideration, we feel that it has merit but does not fully meet PLOS ONE’s publication criteria as it currently stands. Therefore, we invite you to submit a revised version of the manuscript that addresses the points raised during the review process.

We look forward to receiving your revised manuscript.

Kind regards,

Sreeram V. Ramagopalan

Academic Editor

PLOS ONE

“We thank the REWHARD consortium (supported by the Swedish Research Council, grant number 2017-00624) for the data utilized.”

“This work was supported by unrestricted grants from the Swedish Social Insurance Agency (https://www.forsakringskassan.se/om-forsakringskassan/kunskap-och-forskning; grant number: 008430-2019; awarded to EF). The funder had no role in study design, data collection and analysis, decision to publish, or preparation of the manuscript.”

“I have read the journal's policy and the authors of this manuscript have the following competing interests: AM and EF were partly funded by research grants from Biogen. EF has received unrestricted researcher-initiated grants from Bristol-Myers Squibb/Celgene. KA has received unrestricted researcher-initiated grants from Biogen. JH received honoraria for serving on advisory boards for Biogen and Novartis and speaker’s fees from Biogen, Merck-Serono, Bayer-Schering, Teva, and Sanofi-Aventis. He has served as P.I. for projects sponsored by or received unrestricted research support from, Biogen, Merck-Serono, TEVA, Novartis, and Bayer-Schering. JH’s MS research is also funded by the Swedish Research Council. Authors AA and EP have non-financial interests to disclose.”

6. We note that you have indicated that data from this study are available upon request. PLOS only allows data to be available upon request if there are legal or ethical restrictions on sharing data publicly. For more information on unacceptable data access restrictions, please see http://journals.plos.org/plosone/s/data-availability#loc-unacceptable-data-access-restrictions.

7. We note that you have included the phrase “data not shown” in your manuscript. Unfortunately, this does not meet our data sharing requirements. PLOS does not permit references to inaccessible data. We require that authors provide all relevant data within the paper, Supporting Information files, or in an acceptable, public repository. Please add a citation to support this phrase or upload the data that corresponds with these findings to a stable repository (such as Figshare or Dryad) and provide and URLs, DOIs, or accession numbers that may be used to access these data. Or, if the data are not a core part of the research being presented in your study, we ask that you remove the phrase that refers to these data.

Reviewers' comments:

Reviewer's Responses to Questions

**Comments to the Author**

1. Is the manuscript technically sound, and do the data support the conclusions?

Reviewer #1: Yes

2. Has the statistical analysis been performed appropriately and rigorously? 

Reviewer #1: Yes

3. Have the authors made all data underlying the findings in their manuscript fully available?

Reviewer #1: Yes

4. Is the manuscript presented in an intelligible fashion and written in standard English?

Reviewer #1: Yes

5. Review Comments to the Author

Reviewer #1: Thank you to the authors for this interesting piece of research. This paper aimed to determine the relationship between earnings and gender for those diagnosed with MS. The analysis also analysed the interactions between gender, MS occupation, family status. Comparisons were made between those without MS across occupation type and family status. The study was carried out through a population matching cohort analysis.

The findings are inline with previous studies and the additional deeper analysis of gender and MS provides valuable insight into this important topic.

The introduction provided good background to published studies on gender inequalities, the impact of earnings following a diagnosis of MS and the theorised reasons for these variations.

The methods are robust and the authors have used additional techniques to allow for some of the limitations of the data (ie using Huber's).

There are a number of different results to consider due to the main effect and interactions factorial results. These are well explained by the authors and the final section of the results does a great job at summarising a complex analysis.

The discussion is well written and insightful. I particularly felt it was important to discuss the impact of part time hours and the author's discussion of methods used to try and limit this (through SA/DP) is welcomed.

Overall I found this paper very useful and enjoyed reading it. I have set out one minor comment for the authors consideration below:

Abstract/Introduction: (Line 49 and Line 104) The authors are looking at a number of relationships for earnings in the paper, I wonder if it would be helpful to the reader to have the aims of the study to be even more explicit. E.g. that it is looking at gender differences between those diagnosed MS and those without MS AND between those diagnosed while also considering family status and occupation.

6. PLOS authors have the option to publish the peer review history of their article (what does this mean?). If published, this will include your full peer review and any attached files.

Reviewer #1: No

---

## [Author Response · Author response to Decision Letter 0]

30 Jun 2023

RESPONSE LETTER

Manuscript PONE-D-22-32848

Dear Editor,

Thank you for the possibility to submit a revised version of our manuscript titled "Gender differences in earnings among people with multiple sclerosis and associations with type of occupation and family composition: a population-based study with matched references". We also thank the reviewer for the valuable comments. We have carefully addressed the issue raised. Please, find below our point-by-point responses. In addition, changes have been made according to the comments and tracked in the revised manuscript, including reference list formating, using the Track-Changes tool. 

On behalf of all authors, yours sincerely,

Alejandra Machado,

Corresponding author

 

Reviewer Comments 

Reviewer's Responses to Questions

1. Is the manuscript technically sound, and do the data support the conclusions?

Reviewer #1: Yes

2. Has the statistical analysis been performed appropriately and rigorously? 

Reviewer #1: Yes

3. Have the authors made all data underlying the findings in their manuscript fully available?

Reviewer #1: Yes

4. Is the manuscript presented in an intelligible fashion and written in standard English?

Reviewer #1: Yes

 

Comments to the Author

Reviewer #1: Thank you to the authors for this interesting piece of research. This paper aimed to determine the relationship between earnings and gender for those diagnosed with MS. The analysis also analysed the interactions between gender, MS occupation, family status. Comparisons were made between those without MS across occupation type and family status. The study was carried out through a population matching cohort analysis.

The findings are in line with previous studies and the additional deeper analysis of gender and MS provides valuable insight into this important topic.

The introduction provided good background to published studies on gender inequalities, the impact of earnings following a diagnosis of MS and the theorised reasons for these variations.

The methods are robust and the authors have used additional techniques to allow for some of the limitations of the data (ie using Huber's).

There are a number of different results to consider due to the main effect and interactions factorial results. These are well explained by the authors and the final section of the results does a great job at summarising a complex analysis.

The discussion is well written and insightful. I particularly felt it was important to discuss the impact of part time hours and the author's discussion of methods used to try and limit this (through SA/DP) is welcomed.

Overall I found this paper very useful and enjoyed reading it. I have set out one minor comment for the authors consideration below:

Abstract/Introduction: (Line 49 and Line 104) The authors are looking at a number of relationships for earnings in the paper, I wonder if it would be helpful to the reader to have the aims of the study to be even more explicit. E.g. that it is looking at gender differences between those diagnosed MS and those without MS AND between those diagnosed while also considering family status and occupation.

Authors response: Thank you for your comments and the opportunity to further clarify the aims of the study. We have now stated explicitly that differences are observed between those with and without MS while considering type of occupation and family composition among PwMS. (See: Page 2; lines 49-50; and Page 4, lines 104-105).

---

## [Editor Report · Decision Letter 1]

10 Jul 2023

Gender differences in earnings among people with multiple sclerosis and associations with type of occupation and family composition: a population-based study with matched references

PONE-D-22-32848R1

Dear Dr. Machado,

We’re pleased to inform you that your manuscript has been judged scientifically suitable for publication and will be formally accepted for publication once it meets all outstanding technical requirements.

Kind regards,

Sreeram V. Ramagopalan

Academic Editor

PLOS ONE
---

## [Editor Report · Acceptance letter]

12 Jul 2023

PONE-D-22-32848R1 

Gender differences in earnings among people with multiple sclerosis and associations with type of occupation and family composition: a population-based study with matched references 

Dear Dr. Machado:

I'm pleased to inform you that your manuscript has been deemed suitable for publication in PLOS ONE. Congratulations! Your manuscript is now with our production department. 

Kind regards, 

on behalf of

Dr. Sreeram V. Ramagopalan 

Academic Editor

PLOS ONE